# Functional Neural Networks for Parametric Image Restoration Problems

**Fangzhou Luo**
McMaster University
luof1@mcmaster.ca

**Xiaolin Wu**[*]
McMaster University
xwu@ece.mcmaster.ca

**Yanhui Guo**
McMaster University
guoy143@mcmaster.ca

## Abstract

Almost every single image restoration problem has a closely related parameter, such as the scale factor in super-resolution, the noise level in image denoising, and the quality factor in JPEG deblocking. Although recent studies on image restoration problems have achieved great success due to the development of deep neural networks, they handle the parameter involved in an unsophisticated way. Most previous researchers either treat problems with different parameter levels as independent tasks, and train a specific model for each parameter level; or simply ignore the parameter, and train a single model for all parameter levels. The two popular approaches have their own shortcomings. The former is inefficient in computing and the latter is ineffective in performance. In this work, we propose a novel system called functional neural network (FuncNet) to solve a parametric image restoration problem with a single model. Unlike a plain neural network, the smallest conceptual element of our FuncNet is no longer a floating-point variable, but a function of the parameter of the problem. This feature makes it both efficient and effective for a parametric problem. We apply FuncNet to super-resolution, image denoising, and JPEG deblocking. The experimental results show the superiority of our FuncNet on all three parametric image restoration tasks over the state of the arts.

## 1 Introduction

Image restoration [40] is a classical yet still active topic in low-level computer vision, which estimates the original image from a degraded measurement. For example, single image super-resolution [16] estimates the high-resolution image from a downsampled one, image denoising [4] estimates the clean image from a noisy one, and JPEG deblocking [7] estimates the original image from a compressed one. It is a challenging ill-posed inverse problem which aims to recover the information lost to the image degradation process [3], and it is also important since it is an essential step in various image processing and computer vision applications [71, 58, 2, 33, 46, 31, 20].

Almost every single image restoration problem has a closely related parameter, such as the scale factor in super-resolution, the noise level in image denoising, and the quality factor in JPEG deblocking. The parameter in an image restoration problem tends to have a strong connection with the image degradation process. In the super-resolution problem, the blur kernel of the downsampling process is determined by the scale factor [62]. In the image denoising problem, the standard deviation of the additive white Gaussian noise is determined by the noise level [11]. In the JPEG deblocking problem, the quantization table for DCT coefficients is determined by the quality factor [42]. When we try to restore the clean image from a corrupt one, we might know the value of the corresponding parameter for various reasons. In the super-resolution problem, the scale factor is specified by users [28]. In the image denoising problem, the noise level could be measured by other devices [43, 37]. In the

---

[*]Corresponding author

35th Conference on Neural Information Processing Systems (NeurIPS 2021).

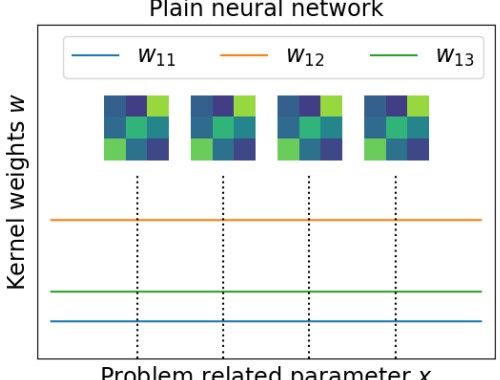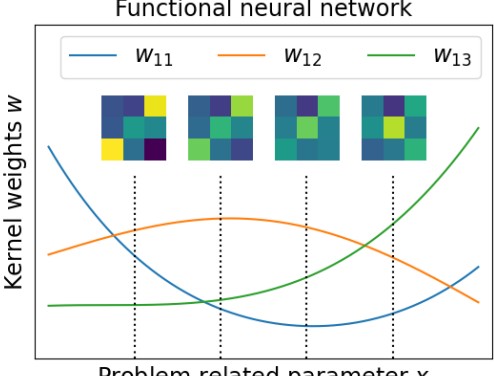

Figure 1: The difference between a plain neural network and our functional neural network (FuncNet). The left and right figure visualize a $3 \times 3$ convolution kernel in a plain neural network and its counterpart in a FuncNet respectively. For the kernel in a plain network, its weights remain unchanged for different problem related parameter levels, so the network only has a limited adaptability to parametric image restoration problems. Unlike a plain network, the smallest conceptual element of our FuncNet is no longer a floating-point variable, but a function of the problem related parameter. In other words, the kernel weights of our FuncNet can change for different situations and make our FuncNet perform better for parametric image restoration problems.

JPEG deblocking problem, the quality factor could be derived from the header of the JPEG file [10]. Therefore, it is very important to use the known parameter well in such a parametric image restoration problem.

Recently, deep convolutional neural network based methods are widely used to tackle the image restoration tasks, including super-resolution [14, 28, 45, 34, 47, 50, 68, 67, 12, 41], image denoising [25, 5, 64, 32, 48, 63, 65, 60, 1], and JPEG deblocking [36, 13, 55, 18, 6, 17, 35, 66, 15]. They have achieved significant improvements over conventional image restoration methods due to their powerful learning ability. However, they have not been paying attention to the parameter involved in an image restoration problem, and handled it in an unsophisticated way. Most previous researchers either treat problems with different parameter levels as independent tasks, and train a specific model for each parameter level [14, 45, 67, 63, 66]; or simply ignore the parameter, and train a single model for all parameter levels [28, 60]. The two popular approaches have their own shortcomings. The former is inefficient in computing, because they may have to train and store dozens of models for different parameter levels. The latter is ineffective in performance, since they ignore some important information that could have helped the restoration process.

To overcome these weaknesses, we propose a novel system called functional neural network (FuncNet) to solve a parametric image restoration problem with a single model. The difference between a plain neural network and our FuncNet is shown in Figure 1. Unlike a plain neural network, the smallest conceptual element of our FuncNet is no longer a floating-point weight, but a function of the parameter of the problem. When we train a FuncNet, we gradually change these functions to reduce the value of the loss function. When we use the trained FuncNet to restore a degraded image with a given parameter, we first evaluate those functions with the parameter, use the evaluation values as weights of the network, and then do inference as normal. This feature makes it both efficient and effective for a parametric problem. By this way, we neatly blend the parameter information into a neural network model, use it to help the restoration process, and only increase a negligible amount of computation as we will demonstrate later. We apply FuncNet to super-resolution, image denoising, and JPEG deblocking. The experimental results show the superiority of our FuncNet on all three parametric image restoration tasks over the state of the arts.

The remainder of the paper is organized as follows. Section 2 provides a brief survey of related work. Section 3 presents the proposed FuncNet model, discusses the details of implementation, and analyses its storage and computational efficiency. In Section 4, extensive experiments are conducted to evaluate FuncNets on three parametric image restoration tasks. Section 5 concludes the paper.

## 2 Related Work

**Neural networks for parametric problems.** To the best of our knowledge, there are seven ways to solve a parametric problem with neural network based methods. We list them below roughly in the order of popularity, and discuss their advantages and disadvantages.

The first method treats problems with different parameter levels as independent tasks, and trains a specific model for each parameter level [14, 45, 67, 63, 66]. The overwhelming majority of previous papers use this approach. This method is easy to understand, and generally has good performance since the parameter level is implied in a model. But it is inefficient in computing, because we may have to train and store dozens of models for different parameter levels.

The second method simply ignores the parameter, and trains a single model for all parameter levels [28, 60]. This method is also easy to understand, and is very efficient since we only need to train and store a single model. But its performance is typically lower than the first method, since we ignore some important information that could have helped the restoration process.

The third method trains a model with a shared backbone for all parameter levels and multiple specific blocks for each parameter level [34]. It is a compromise between the first and the second method, and has acceptable performance and efficiency. But we may still have to store dozens of specific blocks for different parameter levels. And if the capacity of a specific block is not large enough, the block cannot take full advantage of the parameter information.

The fourth method converts the parameter scalar into a parameter map, and treats the map as an additional channel of the input degraded image [65]. It is another way to blend the parameter information into a neural network model. However, the performance of this method is only marginally higher than the second method, and it is still not as good as the first method. Due to the huge semantic difference between the corrupt image and the parameter map, it is hard to make much use of the parameter information for the model.

The fifth method conditions a network by modulating all its intermediate features by scalar parameters [9, 21] or maps [52]. It is another way to make a network adapt to different situations. But unlike our FuncNet, the method changes only features rather than parameters.

The sixth method trains a model with a relatively shallow backbone network, and each filter of the backbone network is generated by a different filter generating network [30, 26, 27, 15]. The filter generating networks are usually multilayer perceptrons, and they take the parameter as input. Since the total size of a model is limited, assigning each filter a unique complex network severely limits the size of the backbone network. Such a shallow backbone network only leads to a mediocre performance. Considering the universal approximation ability of the multilayer perceptron, this method is not that different from training a unique shallow model for each parameter level.

The seventh method searches in the latent space of a generative model, and returns the most probable result which is not contradicting the degradation model with the parameter [57, 38]. It is a general image restoration method which can solve various image restoration problems with a single model, as long as the degradation model is continuously differentiable. However, this method is slower than a feedforward neural network based method, since it requires multiple forward and backward passes in a search. And due to the limited representation ability of the generative model, the performance of this method is also worse than a discriminative model based method.

**Neural network interpolation.** In order to attain a continuous transition between different imagery effects, the neural network interpolation method [54, 53, 51] applies linear interpolation in the parameter space of two trained networks. Although at first glance it is similar to our method, there is a big difference between network interpolation and our FuncNet. The former is a simple interpolation technique while the latter is a regression technique. In the network interpolation method, two CNNs are trained separately for two extreme cases, and then blended in an ad hoc way. This may suffice for tasks [54] and [53], because users will accept roughly characterized visual results, such as "half GAN half MSE" or "half photo half painting". However, this is not good enough for an image restoration task whose goal is to restore the signal as accurately as possible. FuncNet is optimized for the entire value range of the task parameter (e.g., the noise level, SR scale factor), so its accuracy stays high over the entire parameter range, rather than just for the two extreme points like in the network interpolation method.

**Deep CNN for Single Image Super-Resolution.** The first convolutional neural network for single image super-resolution is proposed by Dong et al. [14] called SRCNN, and it achieved superior performance against previous works. Shi et al. [45] firstly proposed a real-time super-resolution algorithm ESPCN by proposing the sub-pixel convolution layer. Lim et al. [34] removed batch normalization layers in the residual blocks, and greatly improved the SR effect. Zhang et al. [67] introduced the residual channel attention to the SR framework. Hu et al. [23] proposed the Meta-Upscale Module to replace the traditional upscale module.

**Deep CNN for Image Denoising.** Zhang et al. [63] proposed DnCNN, a plain denoising CNN method which achieves state-of-the-art denoising performance. They showed that residual learning and batch normalization are particularly useful for the success of denoising. Tai et al. [48] proposed MemNet, a very deep persistent memory network by introducing a memory block to mine persistent memory through an adaptive learning process. Zhang et al. [65] proposed FFDNet, a fast and flexible denoising convolutional neural network, with a tunable noise level map as the input.

**Deep CNN for JPEG deblocking.** Dong et al. [13] proposed ARCNN, a compact and efficient network for seamless attenuation of different compression artifacts. Guo and Chao [18] proposed a highly accurate approach to remove artifacts of JPEG-compressed images, which jointly learned a very deep convolutional network in both DCT and pixel domains. Zhang et al. [66] proposed DMCNN, a Dual-domain Multi-scale CNN to take full advantage of redundancies on both the pixel and DCT domains. Ehrlich et al. [15] proposed a novel architecture which is parameterized by the JPEG files quantization matrix.

# 3 Functional Neural Network (FuncNet)

In this section, we describe the proposed FuncNet model. To transform a plain neural network into a FuncNet, we replace every trainable variable in a plain neural network by a specific function, such as weights and biases in convolution layers or fully connected layers, affine parameters in Batch Normalization layers [24], and slopes in PReLU activation layers [22]; and keep other layers without trainable variables unchanged, such as pooling layers, identity layers, and pixel shuffle layers [45]. We first describe the method to specify the functions in our FuncNet models, then we describe the initialization, training and inference method for these functions, next we describe the network architectures to contain those functions, and finally we analyse the storage and computational efficiency of our FuncNet models.

## 3.1 Specification of Functions

The functions used in our FuncNet model should be simple enough. Let us consider the following failure case as a negative example. Suppose the number of the parameter levels is large but still finite, and we use polynomial functions in our FuncNet. If the polynomial function is too complex, and its degree is greater than or equal to the number of the parameter levels minus one, then our FuncNet is no different from training a specific model for each parameter level. In this case, the failing FuncNet takes as much or even more storage space than multiple independent models, and its inference speed is also slightly slower than a same size plain neural network. This negative example demonstrates the necessity of choosing simple functions for our FuncNet.

We choose the simplest and the most basic kind of function, the linear function, as the functions used in our FuncNet model. In this case, the FuncNet model takes exactly double storage space than a same size plain neural network, and it is still more efficient than storing dozens of models for different parameter levels. And it only increases a negligible amount of computations than a same size plain neural network, as we will demonstrate later. Choosing such a simple function does not lead to a poor performance of the final FuncNet model. With multiple activation layers, the final FuncNet model retains the power of nonlinear fitting. The linear function used in our FuncNet model can be defined as:

$$G(x; \theta_a, \theta_b) = \frac{x - x_a}{x_b - x_a}(\theta_b - \theta_a) + \theta_a \tag{1}$$

where $x$ is the parameter of the problem, $x_a$ and $x_b$ are lower and upper bound of the support of the parameter distribution respectively, $\theta_a$ and $\theta_b$ are trainable variables, and $G(x; \theta_a, \theta_b)$ is the function used in our FuncNet model to generate variables for different parameter levels.

The parameters have different properties for different problems, and we can make the linear function 1 to suit different problems better by replacing $x$ with $H(x)$, where $H(x)$ is a problem-related function. Then the linear function 1 will become:

$$G(H(x); \theta_a, \theta_b) = \frac{H(x) - H(x_a)}{H(x_b) - H(x_a)}(\theta_b - \theta_a) + \theta_a \tag{2}$$

The chosen $H(x)$ should have a physical interpretation related to the problem, and of course should make the final FuncNet model perform well. In the super-resolution problem, we use $H(x) = 1/x$ because the reciprocal of the scale factor is the rescaled length on a low-resolution image from a unit length on a high-resolution image. In the image denoising problem, we use $H(x) = x$ because the noise level is equal to the standard deviation of the additive white Gaussian noise. In the JPEG deblocking problem, we use $H(x) = 5000/x$ for $x \leq 50$ and $H(x) = 200 - 2x$ for $x > 50$. This is the formula used in JPEG standard [42], and it transforms the quality factor into a scale factor of the quantization table for DCT coefficients. The choices of $H(x)$ for different problems are still empirical, just like when people determine the depth, the width or other configuration for a neural network. But we hope that we can determine $H(x)$ automatically in the future, just like what people do in Neural Architecture Search right now [70].

### 3.2 Initialization, Training and Inference

Proper initialization is crucial for training a neural network. Even with modern structures and normalization layers, a bad initialization can still hamper the learning of the highly nonlinear system. The goal of initialization for a plain neural network is to set the value of every trainable variable in a proper range, and to avoid reducing or magnifying the magnitude of input signals exponentially. To properly initialize a function in FuncNet, we have to guarantee that all possible output values of the function are in a proper range. Suppose $H(x)$ in function 2 is a monotonic function, then what we need to do is to use initialization algorithm for a plain neural network [22] to initialize $\theta_a$ and $\theta_b$ independently. In this way, all possible output values of the function 2 lie somewhere between $\theta_a$ and $\theta_b$, and must also be in a proper range if $\theta_a$ and $\theta_b$ are well set. If $\theta_a$ and $\theta_b$ are both sampled from a zero-mean distribution whose standard deviation is $\sigma$, then for all possible output values of the function 2, their expected values are still zero, and their standard deviations are between $\sigma/\sqrt{2}$ and $\sigma$. Experiments have shown that such a small deviation is acceptable for training.

Training a FuncNet is not very different from training a plain neural network. In every iteration, we first sample a fixed number of parameter levels uniformly, use them to construct a minibatch, and then perform stochastic gradient based optimization as normal. As suggested in [69], we train our FuncNet using L1 loss. During the training, we gradually change the values of trainable variables $\theta_a$ and $\theta_b$ to reduce the value of the loss function. The problem can be formulated as

$$\min_{\theta_a, \theta_b} \mathbb{E}_{I^O, I^D, x} \|F(I^D; G(H(x); \theta_a, \theta_b)) - I^O\|_1 \tag{3}$$

where $I^D$ is the degraded version of its original counterpart $I^O$, and $F$ is our FuncNet model.

When we use the trained FuncNet to restore a degraded image with a given parameter $x$, we first evaluate function 2 with $x$, trained $\theta_a$ and $\theta_b$, use the evaluation values as variables of the corresponding plain neural network, and then do inference with the generated plain neural network as normal.

### 3.3 Network Architectures

The requirements of the networks for the three image restoration problems are very different. For the super-resolution problem, the degradation is deterministic and relatively mild, and the network can concentrate on a relatively small area. For the image denoising problem, the degradation is random and relatively severe, and the network needs to pay attention to a larger area. For the JPEG deblocking problem, the degradation occurs in the DCT domain, and the network should have the ability to utilize the information in the DCT domain. So we use individually designed architectures for the three image restoration problems to meet their own requirements.

We directly use architectures of the state-of-the-art plain neural networks for the three tasks as the architectures of our FuncNet models, and we only make essential modifications to them. For the

super-resolution problem, we use the architecture of RCAN [67], and replace the upscale module for integer scale factors [45] with the meta upscale module for non-integer scale factors [23]. For the image denoising problem, we apply a modified U-net [44] structure as the backbone and use RCAB [67] as residual blocks. For the JPEG deblocking problem, we use the architecture of DMCNN [66] for reference. For the DCT domain branch of our JPEG deblocking model, we use frequency component rearrangement to get a more meaningful DCT representation as suggested in [15]; and for the pixel domain branch, we use the same architecture of our image denoising model. More detailed information can be found in the supplementary material.

### 3.4 Storage and Computational Efficiency Analysis

Our FuncNet models have high storage efficiency. As we described in Section 3.1, we use two-degree-of-freedom functions in FuncNet models. This takes only twice as much space as what a plain neural network with the same architecture will take. So storing a FuncNet model is much cheaper than storing dozens of plain networks for different parameter levels. Take the super-resolution task as an example. Suppose the scale factor varies from 1.1 to 4 with stride 0.1 as suggested in [23], we can save 93.3% on storage space by using FuncNet rather than plain neural networks.

Our FuncNet models have high computational efficiency as well. For the training phase, we only need to train one FuncNet model rather than to train dozens of plain networks individually. The computational efficiency analysis for this phase is similar to the preceding storage efficiency analysis. For the inference phase, as we described in Section 3.2, we first evaluate functions with the problem related parameter, use the evaluation values as variables of the corresponding plain neural network, and then do inference with the generated plain neural network as normal. So compared to a plain neural network with the same architecture, our FuncNet model only needs a little extra effort to evaluate functions. This part of computation is directly proportional to the number of parameters in the corresponding plain network, and it is several orders of magnitude smaller than the number of multi-adds for a plain image restoration network. Still take the super-resolution task as an example. Suppose we need to double the size of a 360p image, our FuncNet model only needs extra 0.0001% computation than a plain neural network with the same architecture.

## 4 Experiments

### 4.1 Training Settings

**Training datasets.** Following [67, 23, 60], we use the DIV2K dataset [49] for training. There are 1000 high-quality images in the DIV2K dataset, 800 images for training, 100 images for validation and 100 images for testing. All our three FuncNet models for the three parametric image restoration tasks are trained with the DIV2K training images set.

**Parametric settings.** In all three parametric problems, the problem related parameters are sampled uniformly. In the super-resolution problem, the training scale factors vary from 1.1 to 4 with stride 0.1. In the image denoising problem, the training noise levels are sampled from the uniform distribution on the interval (0, 75). In the JPEG deblocking problem, the quality factors vary from 10 to 80 with stride 2.

**Degradation models.** In the super-resolution problem, we use the bicubic interpolation by adopting the Matlab function imresize to simulate the LR images. In the image denoising problem, we generate the additive white Gaussian noise dynamically by using the Numpy function. In the JPEG deblocking problem, we use the Matlab JPEG encoder to generate the JPEG images.

**Data augmentations.** In all three parametric problems, we use the same data augmentation method. We randomly augment the image patches by flipping horizontally, flipping vertically and rotating $90°$.

**Optimization settings.** In the super-resolution problem, we randomly extract 32 LR RGB patches with the size of $40 \times 40$ as a batch input. In the image denoising problem, we randomly extract 32 RGB patches with the size of $96 \times 96$ as a batch input. In the JPEG deblocking problem, we randomly extract 32 gray patches with the size of $96 \times 96$ as a batch input, and we make sure that the image patches are aligned with boundaries of Minimum Coded Unit blocks. All our three FuncNet models are trained by ADAM optimizor with $\beta_1 = 0.9$, $\beta_2 = 0.999$, and $\epsilon = 10^{-8}$. The initial learning

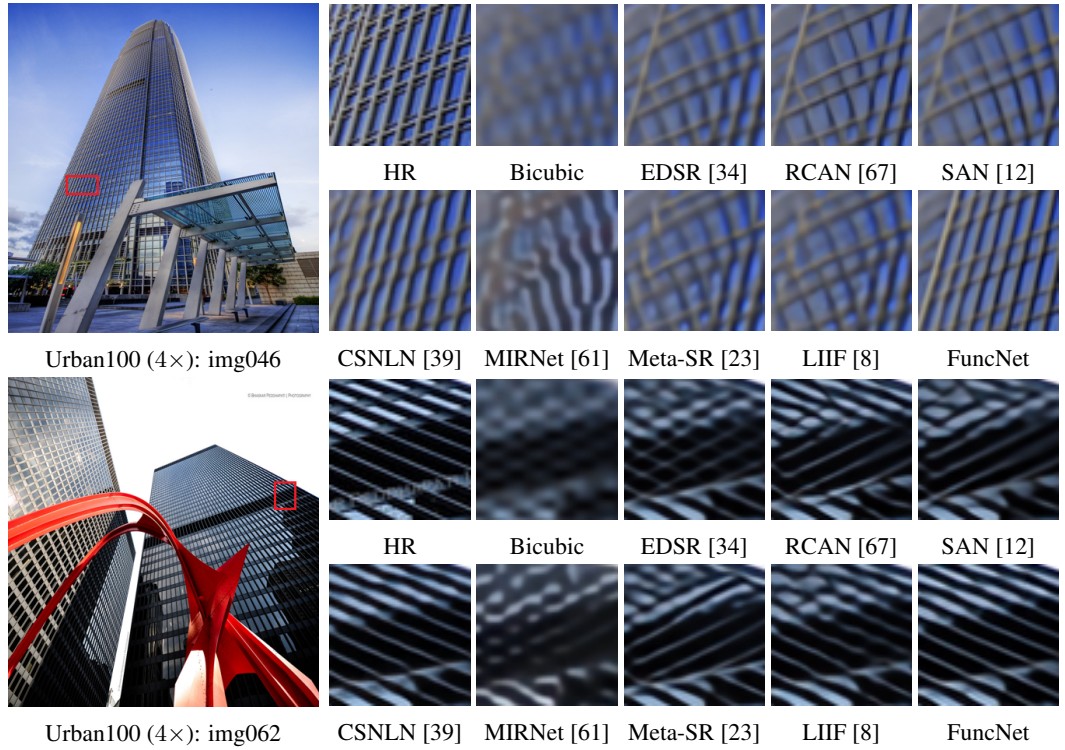

Figure 2: Visual comparison between different super-resolution methods

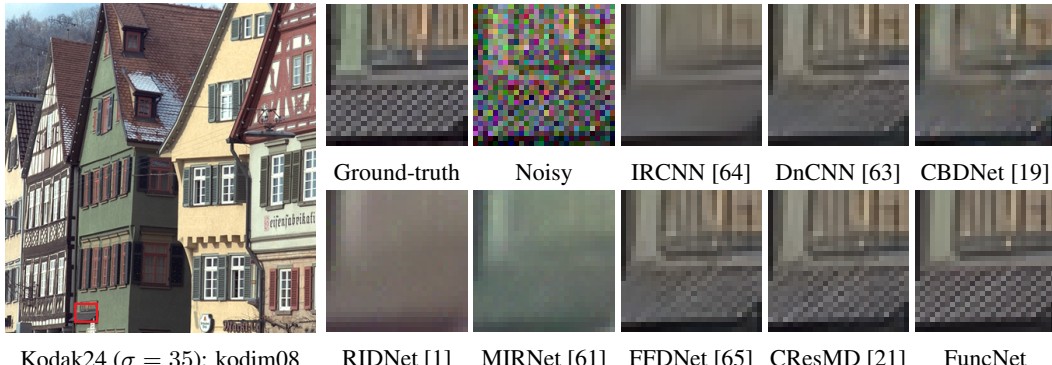

Figure 3: Visual comparison between different image denoising methods

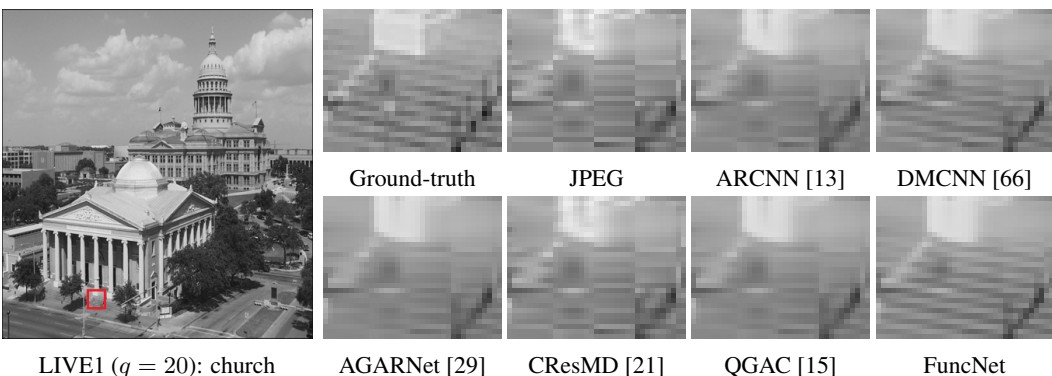

Figure 4: Visual comparison between different JPEG deblocking methods

Table 1: Results of decimal upscale SR on B100. Best and second best results are **highlighted** and underlined

| Scale / Method | ×1.1 | ×1.2 | ×1.3 | ×1.4 | ×1.5 | ×1.6 | ×1.7 | ×1.8 | ×1.9 |
|---|---|---|---|---|---|---|---|---|---|
| Meta-SR [23] | 42.82 | 40.04 | 38.28 | 36.95 | 35.86 | 34.90 | 34.13 | 33.45 | 32.86 |
| FuncNet | **43.43** | 40.41 | 38.55 | 37.16 | 36.02 | 35.08 | 34.26 | 33.60 | 32.98 |
| FuncNet+ | 43.36 | **40.46** | **38.59** | **37.21** | **36.06** | **35.12** | **34.30** | **33.64** | **33.02** |

| Scale / Method | ×2.1 | ×2.2 | ×2.3 | ×2.4 | ×2.5 | ×2.6 | ×2.7 | ×2.8 | ×2.9 |
|---|---|---|---|---|---|---|---|---|---|
| Meta-SR [23] | 31.82 | 31.41 | 31.06 | 30.62 | 30.45 | 30.13 | 29.82 | 29.67 | 29.40 |
| FuncNet | 31.99 | 31.59 | 31.23 | 30.87 | 30.58 | 30.30 | 30.05 | 29.77 | 29.59 |
| FuncNet+ | **32.02** | **31.62** | **31.26** | **30.90** | **30.62** | **30.34** | **30.09** | **29.81** | **29.63** |

| Scale / Method | ×3.1 | ×3.2 | ×3.3 | ×3.4 | ×3.5 | ×3.6 | ×3.7 | ×3.8 | ×3.9 |
|---|---|---|---|---|---|---|---|---|---|
| Meta-SR [23] | 28.87 | 28.79 | 28.68 | 28.54 | 28.32 | 28.27 | 28.04 | 27.92 | 27.82 |
| FuncNet | 29.17 | 29.02 | 28.81 | 28.62 | 28.46 | 28.34 | 28.21 | 28.06 | 27.93 |
| FuncNet+ | **29.20** | **29.06** | **28.85** | **28.67** | **28.51** | **28.38** | **28.26** | **28.11** | **27.97** |

Table 2: Results of integer upscale SR. Best and second best results are **highlighted** and underlined. B100, Urban and Manga represent datasets B100, Urban100, and Manga109 respectively.

| Method | Scale = 2 | | | Scale = 3 | | | Scale = 4 | | |
|---|---|---|---|---|---|---|---|---|---|
| | B100 | Urban | Manga | B100 | Urban | Manga | B100 | Urban | Manga |
| EDSR [34] | 32.32 | 32.93 | 39.10 | 29.25 | 28.80 | 34.17 | 27.71 | 26.64 | 31.02 |
| RCAN [67] | 32.41 | 33.34 | 39.44 | 29.32 | 29.09 | 34.44 | 27.77 | 26.82 | 31.22 |
| SAN [12] | 32.42 | 33.10 | 39.32 | 29.33 | 28.93 | 34.30 | 27.78 | 26.79 | 31.18 |
| CSNLN [39] | 32.40 | 33.25 | 39.37 | 29.33 | 29.13 | 34.45 | 27.80 | 27.22 | 31.43 |
| MIRNet [61] | - | - | - | 27.04 | 24.53 | 26.99 | 25.96 | 23.24 | 25.50 |
| Meta-SR [23] | 32.35 | - | 39.18 | 29.30 | - | 34.14 | 27.75 | - | 31.03 |
| LIIF [8] | 32.32 | 32.87 | - | 29.26 | 28.82 | - | 27.74 | 26.68 | - |
| FuncNet | 32.48 | 33.61 | 39.73 | 29.39 | 29.42 | 34.90 | 27.87 | 27.15 | 31.71 |
| FuncNet+ | **32.51** | **33.78** | **39.87** | **29.43** | **29.57** | **35.10** | **27.90** | **27.29** | **31.97** |

Table 3: Results of image denoising. Best and second best results are **highlighted** and underlined. CBSD, Kodak and Mac represent datasets CBSD68, Kodak24 and McMaster respectively.

| Method | $\sigma = 15$ | | | $\sigma = 35$ | | | $\sigma = 75$ | | |
|---|---|---|---|---|---|---|---|---|---|
| | CBSD | Kodak | Mac | CBSD | Kodak | Mac | CBSD | Kodak | Mac |
| DnCNN [63] | 33.89 | 34.48 | 33.44 | 29.58 | 30.46 | 30.14 | 24.47 | 25.04 | 25.10 |
| CBDNet [19] | 32.67 | 33.32 | 32.87 | 28.11 | 28.87 | 28.77 | 24.05 | 24.64 | 24.38 |
| MIRNet [61] | 27.44 | 28.30 | 27.92 | 22.39 | 23.19 | 22.47 | 18.77 | 18.88 | 18.76 |
| FFDNet [65] | 33.87 | 34.63 | 34.66 | 29.58 | 30.57 | 30.81 | 26.24 | 27.27 | 27.33 |
| CResMD [21] | 33.97 | 34.80 | 34.80 | 29.70 | 30.75 | 31.00 | 26.26 | 27.36 | 27.39 |
| FuncNet | 34.26 | 35.21 | 35.39 | 30.02 | 31.24 | 31.61 | 26.72 | 27.98 | 28.18 |
| FuncNet+ | **34.28** | **35.25** | **35.44** | **30.05** | **31.29** | **31.67** | **26.76** | **28.05** | **28.26** |

Table 4: Results of JPEG deblocking. Best and second best results are **highlighted** and underlined. LIVE and BSDS represent datasets LIVE1 and BSDS500 respectively.

| Method | Quality = 10 | | Quality = 20 | | Quality = 30 | | Quality = 40 | |
|---|---|---|---|---|---|---|---|---|
| | LIVE | BSDS | LIVE | BSDS | LIVE | BSDS | LIVE | BSDS |
| ARCNN [13] | 29.13 | 29.10 | 31.40 | 31.28 | 32.69 | 32.64 | 33.63 | 33.55 |
| DMCNN [66] | 29.73 | 29.67 | 32.09 | 31.98 | - | - | - | - |
| CResMD [21] | 27.89 | 27.92 | 30.58 | 30.55 | 32.46 | 32.37 | 33.87 | 33.73 |
| QGAC [15] | 29.53 | 29.54 | 31.86 | 31.79 | 33.23 | 33.12 | - | - |
| FuncNet | 29.77 | 29.68 | 32.20 | 32.05 | 33.63 | 33.44 | 34.63 | 34.41 |
| FuncNet+ | **29.81** | **29.71** | **32.23** | **32.07** | **33.66** | **33.47** | **34.66** | **34.44** |

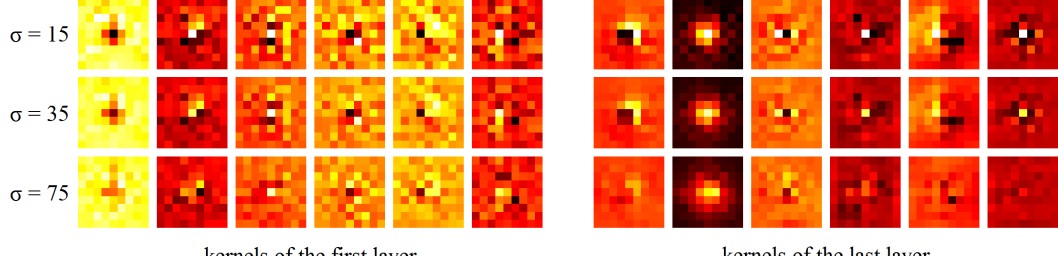

|  σ = 15 |
|  σ = 35 |
|  σ = 75 |

kernels of the first layer                    kernels of the last layer

Figure 5: Kernel visualization of the denoising FuncNet. The left part is sampled from the first layer and the right part is sampled from the last layer. We can find out that the FuncNet uses more radical kernels when the noise level is low, and uses more moderate kernels when the noise level is high.

rate is set to $10^{-4}$ and then decreases by half for every $2 \times 10^5$ iterations of back-propagation. All experiments run in parallel on 4 GPUs.

## 4.2 Evaluation on Standard Benchmark Datasets

In the super-resolution problem, we use the B100 dataset for non-integer scale factor testing, and we use five standard benchmark datasets for integer scale factor testing: Set5, Set14, B100, Urban100, and Manga109. The results are evaluated with PSNR and SSIM [56] on Y channel of transformed YCbCr space. In the image denoising problem, we use three standard benchmark datasets: CBSD68, Kodak24, and McMaster. The results are evaluated with PSNR and SSIM [56] on RGB channel as suggested in [63]. In the JPEG deblocking problem, we use two standard benchmark datasets: LIVE1 and BSDS500. The results are evaluated with PSNR, SSIM [56], and PSNR-B [59] on Y channel.

We compare our results with those of state-of-the-art methods for all three parametric problems. Similar to [34], we also apply a self-ensemble strategy to further improve our FuncNet model and denote the self-ensembled one as FuncNet+. The quantitative results are shown in Table 1, 2, 3, and 4. The visual comparisons are shown in Figure 2, 3, and 4. More detailed information can be found in the supplementary material.

## 4.3 Kernel Visualization and Interpretation

We visualize kernels of our FuncNet model and try to understand and interpret them. The key point of the analysis is to find out how kernels change with the problem related parameter. Here we show samples of kernels from the first and the last layer of the denoising FuncNet, since the denoising problem has the most definite physical meaning among the three image restoration problems. The first and the last layer are also easier to understand. The results are shown in Figure 5. We can find out that the FuncNet uses more radical kernels for features when the noise level is low. By doing so, the FuncNet can get more information. And the FuncNet uses more moderate kernels when the noise level is high, so the FuncNet can get less error.

## 4.4 Ablation Study

As we discussed earlier, using functional kernels instead of numerical kernels is the key to making networks perform better for parametric image restoration problems. To verify the effectiveness of our FuncNet models, we train plain counterparts of our FuncNet models, and compare their evaluation results with FuncNets. And to measure the impact of choice on the problem-related function $H(x)$, we train another two versions of FuncNet. The first one always uses the simplest non-trivial mapping $H(x) = x$, and the second one uses a small multilayer perceptron (MLP) with a hidden layer as a universal function approximator for any possible $H(x)$. We then also compare their evaluation results with FuncNet which uses $H(x)$ with a physical interpretation related to the problem. All the networks for ablation study share the same architectures with their corresponding FuncNet models, and all training and evaluation settings remain unchanged. The evaluation results are shown in Table 5.

Table 5: Results of the ablation study. Super-resolution, denoising and deblocking are tested on Urban100, Kodak24 and LIVE1 respectively.

| Method | Super-resolution | | | Denoising | | Deblocking | |
|---|---|---|---|---|---|---|---|
| | $s = 2$ | $s = 3$ | $s = 4$ | $\sigma = 15$ | $\sigma = 35$ | $q = 10$ | $q = 20$ |
| FuncNet | 33.61 | 29.42 | 27.15 | 35.21 | 31.24 | 29.77 | 32.20 |
| Plain net | 33.07 | 28.93 | 26.70 | 34.83 | 30.89 | 29.58 | 31.94 |
| FuncNet ($H(x) = x$) | 33.48 | 29.33 | 27.05 | 35.21 | 31.24 | 29.64 | 32.16 |
| FuncNet ($H$ is a MLP) | 33.60 | 29.38 | 27.02 | 35.19 | 31.20 | 29.69 | 32.17 |

This ablation study shows that the adaptability of our FuncNet model is important for parametric image restoration problems. Once our FuncNet degenerates into a plain network, its adaptability to different parameter levels disappears, and its performance drops remarkably. The results also prove that both identity function and MLP are acceptable choices for $H(x)$. We can simply use those functions for a problem which is hard to design a $H(x)$ with a physical interpretation.

## 5    Conclusions

We propose a novel neural network called FuncNet to solve parametric image restoration problems with a single model. To transform a plain neural network into a FuncNet, all trainable variables in the plain network are replaced by functions of the parameter of the problem. Our FuncNet has both high storage efficiency and high computational efficiency, and the experimental results show the superiority of our FuncNet on three common parametric image restoration tasks over the state of the arts.

## Acknowledgement

This work is supported by the Natural Sciences and Engineering Research Council of Canada.

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
