# OpenReview forum: "Functional Neural Networks for Parametric Image Restoration Problems"
_NeurIPS.cc/2021/Conference — NeurIPS 2021 Poster_

### Official Review · Reviewer_rd8B · 2021-07-08

**Rating:** 6
**Confidence:** 4

**Summary:**

The authors propose a technique to build image restoration networks that are robust to variations in the problem parameters (e.g. noise level or upscaling factor). The main idea is to derive the weights of the network as a learnable function of the problem parameter.

**Limitations And Societal Impact:**

Societal impact is not discussed.

**Main Review:**

Overall, the paper is well written and the idea clearly explained. The issue of making models that gracefully handle multiple values of a problem parameter is important and I agree with the authors that has been mostly overlooking or treated with unsophisticated techniques. Some works in real image denoising include specialized techniques to estimate the noise level and therefore be robust to variations but the proposed technique is much more general and applicable to different problems. The idea is original and, while simple, elegantly addresses the problem.

The experimental setting is not very clear. What is the setting used for the baselines? Is a baseline model trained for each parameter value or is a single model trained blindly for all values? Ideally, I think both scenarios should be shown because models typically perform better in the former, although, as the authors say, the storage requirements multiply times the number of parameter configurations.  In the blind case, instead, there would be a problem of fairness of the results because the proposed technique effectively uses twice as many trainable parameters with respect to the baseline and it is possible that the improved performance is due to that rather than the technique. The authors should clarify these points.

**Time Spent Reviewing:**

1

---

> ### Author Response · Authors · 2021-08-10
> **To Reviewer rd8B**
>
>
> Comment: What is the setting used for the baselines? Is a baseline model trained for each parameter value or is a single model trained blindly for all values?
>
> Response: For all other methods, we use their official code and models for comparison. For most of them, their training procedures are like ours, with the same training set DIV2K, the same loss function, the same optimizer and similar learning rate. The only major difference is the way to treat different parameter levels. For example, DnCNN officially trained a specific model for each noise level, while FFDNet officially trained a single model for all noise levels. We did not change their training strategy for different parameter levels, so some of them are trained with a specific parameter level, and others are trained with varied parameter levels. We will clarify this in Section 4.2.
>
> Comment: In the blind case, instead, there would be a problem of fairness of the results because the proposed technique effectively uses twice as many trainable parameters with respect to the baseline and it is possible that the improved performance is due to that rather than the technique.
>
> Response: Good point. We will add this comparison in the revised version of our paper. However, we believe that this comparison is not so fair for us. As we discussed in Section 3.4, although FuncNet takes twice as much space as a plain neural network of the same architecture, it only needs a negligible amount of extra computation. If we double the size of the plain model of the same architecture, its multiply–accumulate operations will be nearly twice as much as the proposed FuncNet.

---

### Official Review · Reviewer_rgzb · 2021-07-14

**Rating:** 6
**Confidence:** 4

**Summary:**

This paper introduces functional neural networks for image restoration that are parametrized by a variable (e.g. super-resolution factor, noise level, JPEG quality level).  A functional network is one where the neural network parameters themselves are a function of the problem parameter known a priori.  This way, the neural network weights can adapt based on the problem, for example, a denoising CNN can apply stronger denoising when the noise level increases.  Different functions are explored including linear functions which require twice the storage but using a single model.  Compelling experimental results are shown across a range of parameters of the problem.

**Ethical Concerns:**

None are identified

**Limitations And Societal Impact:**

The paper appears to be missing a societal impact statement.  The paper also doesn’t really characterise its limitations.

Suggestions would to be prepare a societal impact statement.  For limitations, it would be helpful to understand if the network can become unstable as mentioned above when training.  It would be helpful to see some examples where the method fails to produce an excellent result to fully characterise the method.

**Main Review:**

The paper addresses an important and common problem in image restoration – how to develop a single neural network that can work effectively across a range of settings (e.g. a denoiser that can denoise images with varying levels of noise).  The paper effectively argues that making neural networks for each setting produces good results, but also many networks; whereas a single network that works across a large range of settings may be less effective.  Indeed this a well-known issue.

The paper doesn’t use this terminology, but essentially it is proposing a solution for a conditional neural network.  That is, the network itself is conditioned based on the parameter (e.g. noise level).  The paper provides a nice taxonomy of different approaches to designing conditional neural networks, categorized into six methods in Section 2.  However, this taxonomy is incomplete; there are other methods proposed for conditional networks that appear to be outside the six methods described, for example:

•	Wang et al., “Recovering Realistic Texture in Image Super-resolution by Deep Spatial Feature Transform,” CVPR 2018 introduces Spatial Feature Transform modules which modulate the features, conditioned in their case on a segmentation map but this could instead be a map representing the a priori parameter of the problem.

•	Choi et al., “Variable Rate Deep Image Compression With a Conditional Autoencoder,” ICCV 2019 introduce a conditioning mechanism using a Lagrange multiplier to produce a conditional autoencoder to avoid training multiple networks in a similar way to the proposed paper.

•	He et al., "Interactive Multi-Dimension Modulation with Dynamic Controllable Residual Learning for Image Restoration," ECCV 2020 uses a conditional network to control residual connections.

Therefore, the paper may benefit from recasting the problem as one of learning a conditional neural network for image restoration.  The approach of modifying the network parameters rather than the inputs or features appears to be novel.

The approach to adjusting the neural network weights based on the parameter is simple and allows the network parameters to change.  Figure 1 gives a nice conceptual visualisation of the concept, and Figure 5 shows the kernel weights being adjusted in practice based on the noise level parameter known a priori.  While this formulation is simple, some details remain unclear to this reviewer.  For example, in practice, how are the lower and upper bounds (x_a, x_b) described below Eq 1 known before the network is trained?  Are there any controls on \theta_a and \theta_b in Equation 1 to ensure that \theta_a < \theta_b?  If this condition wasn’t satisfied it would seem the network would become unstable.  The paper doesn’t comment on this so perhaps it doesn’t happen?  However, these points are unclear to the reader.

The visual results are very impressive, particularly those shown in Figures 2 - 4 which greatly outperform the methods that are compared as there is considerably more detail.  Numerically the results are maybe less pronounced in Tables 2 – 4 however do represent an improvement.  Something worrisome though is the methods used in the comparison.  For denoising, the most recent algorithm is from 2018; for super-resolution 2019.  However, the field is moving fast so it is not convincing that the methods in comparison are truly state-of-the-art; recent papers are including methods like attention for example unlike DnCNN or FFDNet.  One might expect papers from 2020 compared to; or referencing recent challenges at the NTIRE 2020 workshops involving denoising or super-resolution.  For example,

•	Zamir et al., “Learning Enriched Features for Real Image Restoration and Enhancement,” ECCV 2020 has the highest score in 2020 on the paperswithcode leaderboard on SIDD and is used for different image restoration tasks.

The paper has a number of typographical and grammatical issues; please proofread carefully.  There are too many issues to list here.  A common issue is a missing indefinite article, for example, on Line 251, “In super-resolution problem” should be “In the super-resolution problem”.  Note this a common issue in the paper and not limited to Line 251.

Overall, the challenge of conditional image restoration is an important one, and the paper is appreciated by this reviewer.  However, the paper lacks some awareness of the literature and comparisons to other methods are not fully convincing.  If the paper could more thoroughly characterize conditional neural networks and provide more comprehensive comparisons to more recent methods it could be strengthened.

# Update
Thank you for the additional details and clarifications in the author response.  Having read the other reviews and author responses as well, I've raised my rating by one point.  I think this paper is addressing an important problem in an interesting way that has novelty.  I still have concerns about comparisons to more recent work and issues with presentation and grammar which are holding the paper back a bit.

**Time Spent Reviewing:**

4

---

> ### Author Response · Authors · 2021-08-10
> **To Reviewer rgzb**
>
>
> Comment: The paper doesn’t use this terminology, but essentially it is proposing a solution for a conditional neural network. Therefore, the paper may benefit from recasting the problem as one of learning a conditional neural network for image restoration.
>
> Response: We totally agree with you. We will discuss this clearly in the revised version of our paper.
>
> Comment: There are other methods proposed for conditional networks that appear to be outside the six methods described.
>
> Response: Thanks for pointing this out. We will add a new paragraph to discuss the feature modulation approach to condition a network in the related work section.
>
> Comment: In practice, how are the lower and upper bounds (x_a, x_b) described below Eq 1 known before the network is trained?
>
> Response: In this paper, the setting for the lower and upper bounds just follows previous works. But in practice, there are at least two strategies to determine these bounds. The first is to use a reasonably large range of the task parameter. For example, we think [10, 100] is more than enough for real life JPEG deblocking problem. The second strategy is to estimate quantiles from the real distribution of the task parameter, and make the probability that the task parameter is not in (x_a, x_b) negligible.
>
> Comment: Are there any controls on \theta_a and \theta_b in Equation 1 to ensure that \theta_a < \theta_b? If this condition wasn’t satisfied it would seem the network would become unstable. The paper doesn’t comment on this so perhaps it doesn’t happen?
>
> Response: We did not add any constraint on \theta_a and \theta_b. The training of FuncNet converges stably, just like a plain neural network.
>
> Comment: For denoising, the most recent algorithm is from 2018; for super-resolution 2019. However, the field is moving fast so it is not convincing that the methods in comparison are truly state-of-the-art.
>
> Response: Thanks for your suggestion. We will compare our FuncNet with the most recent results in the revised version of our paper.
>
> Comment: The paper has a number of typographical and grammatical issues; please proofread carefully.
>
> Response: We will proofread it thoroughly, and correct the missing indefinite article problem.

---

> ### Author Response · Authors · 2021-09-06
> **To Reviewer rgzb**
>
> Thanks for your feedback. Following your request for comparisons with more recent works, we compare FuncNet with the papers mentioned by reviewers. For all other methods, we use their official code and models in evaluations. Over the entire scale range of the super-resolution task, the average PSNR of FuncNet is 3.27 dB higher than MIRNet [1], 0.37 dB higher than LIIF [2], and 0.13 dB higher than CSNLN [3]. Over the entire range of noise levels for the denoising task, the average PSNR of FuncNet is 0.43 dB higher than CResMD [4], 7.74 dB higher than MIRNet [1], and 2.29 dB higher than CBDNet [5]. For the entire quality range of the JPEG deblocking problem, our PSNR is 1.31 dB higher than CResMD [4] on average. It is very surprising that some recent works perform poorly. Detailed results are attached at the end of the response.
>
> We will also discuss the relation between FuncNet and other conditional neural networks in the revised paper. And we will proofread our paper thoroughly to correct the grammar problems.
>
>
> PSNR results for the super-resolution problem on B100 dataset (x2 / x3 / x4):
>
> MIRNet [1]: ---- / 27.04 / 25.96
>
> LIIF [2]: 32.32 / 29.26 / 27.74
>
> CSNLN [3]: 32.40 / 29.33 / 27.80
>
> FuncNet: 32.48 / 29.39 / 27.87
>
> PSNR results for the super-resolution problem on Urban100 dataset (x2 / x3 / x4):
>
> MIRNet [1]: ---- / 24.53 / 23.24
>
> LIIF [2]: 32.87 / 28.82 / 26.68
>
> CSNLN [3]: 33.25 / 29.13 / 27.22
>
> FuncNet: 33.61 / 29.42 / 27.15
>
> PSNR results for the denoising problem on CBSD68 dataset (sigma = 15 / 35 / 75):
>
> CResMD [4]: 33.97 / 29.70 / 26.26
>
> MIRNet [1]: 27.44 / 22.39 / 18.77
>
> CBDNet [5]: 32.67 / 28.11 / 24.05
>
> FuncNet: 34.26 / 30.02 / 26.72
>
> PSNR results for the denoising problem on Kodak24 dataset (sigma = 15 / 35 / 75):
>
> CResMD [4]: 34.80 / 30.75 / 27.36
>
> MIRNet [1]: 28.30 / 23.19 / 18.88
>
> CBDNet [5]: 33.32 / 28.87 / 24.64
>
> FuncNet: 35.21 / 31.24 / 27.98
>
> PSNR results for the JPEG deblocking problem on LIVE1 dataset (q = 10 / 20 / 30 / 40):
>
> CResMD [4]: 27.89 / 30.58 / 32.46 / 33.87
>
> FuncNet: 29.77 / 32.20 / 33.63 / 34.63
>
> PSNR results for the JPEG deblocking problem on BSDS500 dataset (q = 10 / 20 / 30 / 40):
>
> CResMD [4]: 27.92 / 30.55 / 32.37 / 33.73
>
> FuncNet: 29.68 / 32.05 / 33.44 / 34.41
>
> [1] Learning Enriched Features for Real Image Restoration and Enhancement, ECCV 2020
>
> [2] Learning Continuous Image Representation with Local Implicit Image Function, CVPR 2021
>
> [3] Image Super-Resolution with Cross-Scale Non-Local Attention and Exhaustive Self-Exemplars Mining, CVPR 2020
>
> [4] Interactive Multi-dimension Modulation with Dynamic Controllable Residual Learning for Image Restoration, ECCV 2020
>
> [5] Toward Convolutional Blind Denoising of Real Photographs, CVPR 2019

---

> > ### Comment · Reviewer_rgzb · 2021-09-10
> > **Thank you for the additional experiments**
> >
> > Thank you for the additional experiments, they are encouraging referencing more recent work

---

### Official Review · Reviewer_74Ec · 2021-07-16

**Rating:** 6
**Confidence:** 4

**Summary:**

The authors proposed a functional neural network for image restoration. They use a linear function with respect to the task parameter to adjust the weights of the network, to achieve an adaptive restoration for different parameter levels using a single model. The idea is similar to meta-learning on restoration, or weight modulation derived from instance normalization. The experiment shows its better quantitative performance compared with SOTA among three tasks, and better visual quality.

**Limitations And Societal Impact:**

See the above comments.

**Main Review:**

Pros,

- Elegant design for the proposed functional neural network and the writing of the paper is good and very easy to follow.
- Extensive experiments on SOTA and benchmark dataset revealing the better performance.

Cons,

- The idea is not novel enough, and the function design is not that interesting as in [1]. Besides, the task level is given, so the model is not blind for an adaptive inference with any arbitrary inputs, making the practical values very limited.
- It's better to differentiate task parameter (noise level, scale factor etc.) with network parameter (model weights). Some terms in section 3.1 are somehow confusing, especially 'x' mostly represents input images. When the authors refer x as the parameter, I original though it was the model weights.
- While comparing with other baselines (like denoising or JPEG deblocking), have the author used one single model trained with all the parameter levels, or each model per level separately? Besides, the parameter size should be controlled for a fair comparison. It seems the double parameter size improves the restoration performance. I understand that one set of parameter can represent multiple models with any given x, but it will be more fair to also double the weights of the single model of the same architecture.
- In section 4.3, it's not clear what is the 'input' and 'output' layer the authors refer to. The visualization is a little bit confusing to interpret the results. Since the function is linear, it should not be surprising to see the linear effects of kernels.
- Missing some SR reference like [2]

My main concern of this paper is the novelty and technical levels for NeurIPS, and its practical values in real image restoration. The better performance may be from some unfair comparison, and there is no explicit design for a single model with adjusted weights to be better than others.

[1] Learning Continuous Image Representation with Local Implicit Image Function, CVPR'21
[2] Image Super-Resolution with Cross-Scale Non-Local Attention and Exhaustive Self-Exemplars Mining, CVPR'20



**Time Spent Reviewing:**

0.5

---

> ### Author Response · Authors · 2021-08-10
> **To Reviewer 74Ec**
>
>
> Comment: The idea is not novel enough, and the function design is not that interesting as in [1].
>
> Response: Although [1] also has the word “function” in its title, it is not quite related to our work. It is more like MetaSR, which is the major opponent of [1]. They both train a model with a shared backbone for all scale levels, and use an arbitrary upscale module with their function to replace the traditional integer upscale module. In contrast, FuncNet replaces every trainable variable in a plain neural network by a specific function. And the inputs of their function are features and coordinates, which are not problem-related parameters like ours. We think [1] and MetaSR are both variations of the third method reviewed in Section 2. So, we respectfully disagree with the reviewer that [1] renders our idea not novel enough. Reviewer rgzb and Reviewer rd8B all pointed out, our approach appears to be novel and original.
>
> Comment: The task level is given, so the model is not blind for an adaptive inference with any arbitrary inputs, making the practical values very limited.
>
> Response: We have indeed discussed this in our original paper (Line 33 ~ 38). In practice, we often do know the task parameter values. For super-resolution, the scale factor is specified by the user. For JPEG deblocking, the quality factor is in the header of the JPEG file. For image denoising, the noise level could be estimated by other algorithms, or by a subnet. For the three image restoration tasks in this paper, two of them have the precise task parameter, and the other can have the parameter estimated.  So, our method does have practical value. Actually, one of the motivations of this paper is to make those impractical image restoration neural networks become practical in real life. The overwhelming majority of previous papers treat image restoration problems with different parameter levels as independent tasks, and train a specific model for each parameter level. As such one has to train and store dozens of models, one per parameter level. And the task parameter also needs to be known at inference time, so that people can choose the correct model. Now we only need to train and store one FuncNet model to solve all problem instances.
>
> Comment: It's better to differentiate task parameter (noise level, scale factor etc.) with network parameter (model weights).
>
> Response: Thanks for your good suggestion. We will clarify this in the revised version of our paper.
>
> Comment: While comparing with other baselines (like denoising or JPEG deblocking), have the author used one single model trained with all the parameter levels, or each model per level separately?
>
> Response: For all other methods, we use their official code and models for comparison. For most of them, their training procedures are like ours, with the same training set DIV2K, the same loss function, the same optimizer and similar learning rate. The only major difference is the way to treat different parameter levels. For example, DnCNN officially trained a specific model for each noise level, while FFDNet officially trained a single model for all noise levels. We did not change their training strategy for different parameter levels, so some of them are trained with a specific parameter level, and others are trained with varied parameter levels. We will clarify this in Section 4.2 in revision.
>
> Comment: It will be more fair to also double the weights of the single model of the same architecture for a fair comparison.
>
> Response: Thanks for your suggestion. We will add this comparison in the revised version of our paper. However, we believe that this comparison is not so fair for us. As we discussed in Section 3.4, While FuncNet takes twice as much space as a plain neural network of the same architecture, it only needs a negligible amount of extra computation. If we double the size of the plain model of the same architecture, its multiply–accumulate operations will be nearly twice as much as FuncNet.
>
> Comment: In section 4.3, it's not clear what is the 'input' and 'output' layer the authors refer to. The visualization is a little bit confusing to interpret the results.
>
> Response: You are right. We will use the first/last layer instead of input/output layer in the revised version of our paper. And we will use a better caption for Figure 5. The visualization intends to show that FuncNet uses more radical kernel for features when the noise level is low. By doing so, FuncNet can get more high-frequency details. On the other hand, FuncNet uses more moderate kernels for higher noise levels, so it can denoise better.
>
> Comment: Missing some SR reference like [2]
>
> Response: We will add it in the revised version of our paper.
>
> [1] Learning Continuous Image Representation with Local Implicit Image Function, CVPR'21
>
> [2] Image Super-Resolution with Cross-Scale Non-Local Attention and Exhaustive Self-Exemplars Mining, CVPR'20

---

> > ### Comment · Reviewer_74Ec · 2021-08-31
> > **Thanks**
> >
> > Thanks for the detailed rebuttal.
> >
> > For the novelty, the authors make a fair point. I agree with the authors that the methodology of the mentioned works are different, but the goals are similar. I personally like the methods which are more intuitive. The term 'function' may not be necessarily defined as the function of the parameters as in this paper. In terms of the methodology and implementation, I do believe the proposed methods have the merits, and no previous works (except for the idea of network interpolation) are related to that.
> >
> > It is expected that arbitrary-level SR can also work on different types of downsampling kernels. It will bring better practical values of functional networks. So functions on kernels should be an interesting research directions, and I do recommend the authors to generalize the tasks to deblurring or unknown image restoration. (That's why I thought the models and tasks are not sufficient enough). Those tasks are more 'practical' instead of bicubic downsampling and gaussian denoising. Intuitively, a functional network should have a more powerful ability under blind restoration setting. It is suggested to add more evaluations on that. But the current version may be already sufficient for a publication.
> >
> > Other reviewers also mention the issues of parameter sizes. If the authors claim the double-size network only adds negligible computation to the model, then adding the comparison of computation complexity will be necessary.
> >
> > After seeing the rebuttal and other review comments, I think this paper still has some problems in fair comparison with 'SOTA', but overall it should be sufficient for NeurIPS. I will raise my rating by one point.

---

> > > ### Author Response · Authors · 2021-09-06
> > > **To Reviewer 74Ec**
> > >
> > > Thank you for your constructive feedback. We will study FuncNet for blind deblurring in the future, and the idea of functions on downsampling kernels is really intriguing.
> > >
> > > For the double-size network issue, we are still training those networks for evaluation, and we will add this comparison in the revised paper. As for the computation complexity of FuncNet, we did discuss this in our original paper (Line 214 ~ 225). To resize a 360p image to a 720p image, FuncNet model only needs extra 0.000104% computation than a same-size plain neural network. In contrast, the double-size plain network will spend nearly twice as much computation as FuncNet.

---

### Official Review · Reviewer_hUCx · 2021-07-18

**Rating:** 6
**Confidence:** 5

**Summary:**

This paper proposes a network interpolation scheme for non-blind image restoration
problems with a single parameter (e.g. noise level for denoising).
The parameter is remapped to [0, 1] using an affine mapping.
Effectively, two networks are trained such that
for any given problem parameter in [0,1],
an interpolated network is obtained by linearly blending the two networks
parameters. The interpolated model is then applied to the input.


**Ethical Concerns:**

None of note.

**Limitations And Societal Impact:**

Limitations are not discussed in the paper.

**Main Review:**

The technical contribution of this paper is rather meager, as network interpolation
has been discussed multiple times in the past, including in relation to image
restoration. Still, despite some minor  concerns about the evaluation, the proposed
FuncNet model outperforms reasonable recent baselines, while being trained on
a *range* of parameter values.



### Strengths

- Improving neural network performance on parameterized problem such that
the network adapts to the input parameter is a timely problem with practical
importance.
- Incorporating the parameter as a linear blending weight between two
sets of network parameters (weights and biases) is interesting.
- The proposed method quantitatively outperforms recent previous work
on denoising, super-resolution and JPEG deblocking.


### Weaknesses

- The proposed technical solution is essentially network interpolation, which
is not novel, and should therefore be discussed. For
instance "ESRGAN: Enhanced Super-Resolution Generative Adversarial Networks"
[Wang 2018] linearly blend between the parameters of a model trained with a GAN loss, and one without.
Even closer to the domain discussed here, "Deep Network Interpolation for Continuous Imagery Effect Transition" [Wang 2019]
interpolate between different filter effects.
- How would the method if the restoration task had more than one parameter (e.g. 2 noise parameters for
a Heteroskedastic Gaussian noise model)?
- The discussion of related work is insufficient.
1. For instance, CBDNet "Toward Convolutional Blind Denoising of Real Photographs" [Guo 2019]
uses yet another approach to incorporate the model parameter: an additional prediction branch
supervised to regress the ground-truth parameter.
2. Another largely used approach to condition a network to a scalar parameter,
is to modulate all its intermediate features based on the condition, using a mechanism like
FiLM "FiLM: Visual Reasoning with a General Conditioning Layer" [Perez 2017].
"Flexible Image Denoising with Multi-layer Conditional Feature Modulation" [Du 2020] apply this to denoising.
"Interactive Multi-Dimension Modulation with Dynamic Controllable Residual Learning for Image Restoration" [He 2020]
use a similar conditioning mechanism for image restoration, and deal with multiple parameters simultaneously.
- I have some reserves about the evaluation of the proposed model. Why is RIDNet omitted from
the quantitative tables? Was the model retrained with the same noise model?
FFDNet is a bit outdated for a baseline, consider re-training a more recent
more with the same procedure as the proposed model (i.e. same noise model, etc).
- Thank you for the supplemental material. Seeing more images is useful, but
I would have liked to see more diversity in the input images, rather than
selected crops. It is difficult to judge the robustness of the method otherwise.
- I do not think the self-ensembling (FuncNet+) adds much to the paper.
- It is not clear from the paper whether the baselines were used as-is, retrained
on *all* parameter values (for their respective application), or trained specifically
on each parameter value.
- In general, it would be useful to understand how FuncNet compares to
the "plain" version, specialized to each given parameter. This would give
an upper-bound on the quality the model can reach, and therefore give
a sense of how much quality is lost by using the interpolation scheme instead.



### Misc

- 224: 0.0001% inaccurate, do you rather mean the linear interpolation of parameters
takes this fraction of the total compute?
- The denoising result in Fig3 is nice. I am surprised RIDNet performs so poorly, since it is more
recent, and generally believed to outperform FFDNet.


### Update

I bumped by recommendation by +1. My criticism about the method being "just network interpolation" was not completely fair since the interpolated models are also used and supervised during training. The contribution therefore has some novelty that I did not fully grasp originally.

**Time Spent Reviewing:**

2

---

> ### Author Response · Authors · 2021-08-10
> **To Reviewer hUCx**
>
>
> Comment: The proposed technical solution is essentially network interpolation, which is not novel.
>
> Response: There is a big difference between network interpolation and our FuncNet: the former is a simple interpolation technique while the latter is a regression technique. In the network interpolation method, two CNNs are trained separately for two extreme cases, and then blended in an ad hoc way. This may suffice for tasks [1] and [2], because users will accept roughly characterized visual results, such as "half GAN half MSE" or “half photo half painting”. However, this is not good enough for image restoration task whose goal is to restore the signal as accurately as possible. FuncNet is optimized for the entire value range of the task parameter (e.g., the noise level, SR scale factor), so its accuracy stays high over the entire parameter range, rather than just for the two extreme points like in the network interpolation method. We will discuss the above difference in detail in revision. Thanks for raising the point.
>
> Comment: How would the method if the restoration task had more than one parameter?
>
> Response: FuncNet with multiple parameters is a generalization of the single parameter case. For example, if the problem has two parameters, we can replace the linear blending with bilinear blending for each weight and bias. As we discussed in Section 3.4, this model takes four times as much space as a plain neural network of the same architecture, but it only needs negligible extra computation.
>
> Comment: Another largely used approach to condition a network to a scalar parameter, is to modulate all its intermediate features based on the condition.
>
> Response: Thanks for pointing this out. We will add a new paragraph to discuss the feature modulation approach to condition a network in the related work section. But as Reviewer rgzb pointed out, our approach of modifying the network parameters rather than the inputs or features is novel.
>
> Comment: CBDNet uses yet another approach to incorporate the model parameter.
>
> Response: CBDNet is a variant of FFDNet, which was cited in our paper. It still treats the parameter map as an additional channel of the input degraded image, and the only difference is that the map is estimated rather than given. We will add CBDNet to the related work section, just after FFDNet.
>
> Comment: Why is RIDNet omitted from the quantitative tables?
>
> Response: Sorry, we will add quantitative result of RIDNet in the revised version of our paper. PSNR of our FuncNet is 0.36 ~ 0.61dB higher than PSNR of RIDNet on CBSD68, Kodak24 and McMaster datasets.
>
> Comment: Was the model retrained with the same noise model?
>
> Response: For all other state-of-the-art methods, we use their official code and models for comparison. For most of them, their training procedures are like ours, with the same training set DIV2K, the same loss function, the same optimizer and similar learning rate. The only major difference is the way to treat different parameter values. For example, DnCNN officially trained a specific model for each noise level, while FFDNet officially trained a single model for all noise levels. We did not change their training strategy for different task parameter values, so some of them are trained with a specific parameter value, and others are trained with varied parameter values. In revision, we will clarify this in Section 4.2.
>
> Comment: I do not think the self-ensembling (FuncNet+) adds much to the paper.
>
> Response: The self-ensembling can improve the PSNR by up to 0.26dB (Manga x4) without fine tuning. We think it is useful if we have a powerful computer for inference but do not want retraining.
>
> Comment: 224: 0.0001% inaccurate, do you rather mean the linear interpolation of parameters takes this fraction of the total compute?
>
> Response: Yes, the more accurate number is 0.000104%.
>
> [1] ESRGAN: Enhanced Super-Resolution Generative Adversarial Networks" [Wang 2018]
>
> [2] Deep Network Interpolation for Continuous Imagery Effect Transition" [Wang 2019]

---

> > ### Comment · Reviewer_hUCx · 2021-08-21
> > **Update**
> >
> > Thank you for the clarifications.
> >
> > In particular, I appreciate the clear comparison to network interpolation. I suggest this discussion be added to the paper to alleviate similar confusion in other readers. I believe that adding a network interpolation baseline (i.e. training only for the two extreme values) would strengthen the point.
> >
> > The multi-parameter case seems like a limitation to me. 1. It is unclear how well quality is preserved as the dimension of the n-linear interpolation grows. 2. As you described, the storage requirements grow exponentially, which seems unpractical. I suggest expanding the discussion in the paper.
> >
> > I will upgrade my recommendation based on my new understanding of the first point.

---

> > > ### Author Response · Authors · 2021-08-22
> > > **To Reviewer hUCx**
> > >
> > > Indeed, as of now, without further studies and experiments, we cannot conclude anything about the multi-parameter case. But the storage growth rate is not exponential in the number of task parameters. In the previous reply, we made a mistake to claim that the model takes four times space for two parameters case. This is not correct and we want to clarify it here.
> > >
> > > The multi-parameter generalization of the equation (2) in our paper is
> > >
> > > G(x_1,x_2,...,x_n;theta_0,theta_1,theta_2,...,theta_n) = theta_0 + Sum from i=1 to n (H(x_i)-H(x_ia))/(H(x_ib)-H(x_ia))*(theta_i-theta_0)
> > >
> > > where G(x;theta) is the function used in our FuncNet model to generate variables for different task parameter levels, x_i is the i-th task parameter, x_ia and x_ib are lower and upper bound of the i-th task parameter respectively, theta_i is the i-th trainable network variable, and H(x) is a problem-related function.
> > >
> > > As shown in the above formula, if there are n parameters in the task, we only need n+1 sets of variables. The storage growth rate is still linear in the number of task parameters.

---

> > > > ### Comment · Reviewer_hUCx · 2021-09-10
> > > > **Thank you**
> > > >
> > > > Thank you these clarifications. I have accounted for this information and my original misunderstanding in my final recommendation.

---

### Decision · Program_Chairs · 2021-09-27

**Decision:**

Accept (Poster)

**Comment:**

This paper develops a functional neural network for image restoration/reconstruction tasks. The authors develop an adaptive restoration for different parameter levels using a single model essentially by linearly mixing the two networks. This is somewhat akin to doing meta-learning for restoration. The authors provide various experiments for a variety of restorations tasks showing improvements compared to existing approaches. The reviewers mostly thought the paper's approach was interesting and commended the clarity of the paper. However, they also raised concerns about novelty, differentiation of task and network parameters, baselines. There was a lively discussion in the rebuttal period and both the authors and reviewers engaged in the discussion. As a result most reviewers while somewhat lukewarm lean towards acceptance. I share this opinion. The paper has some interesting ideas but can improve in other aspects. I recommend acceptance but ask the authors to follow the recommendations of the reviewers to improve the final peresentation.